# Readiness of healthcare facilities with tuberculosis services to manage diabetes mellitus in Tanzania: A nationwide analysis for evidence-informed policy-making in high burden settings

Festo K. Shayo[1,2]*, Sigfrid Casmir Shayo[3]

1 Department of Internal Medicine, Muhimbili National Hospital, Dar es Salaam, Tanzania, 2 Muhimbili University of Health and Allied Sciences, Dar es Salaam, Tanzania, 3 Department of Diabetes and Endocrinology, Kagoshima University, Kagoshima, Japan

* festocasmir@gmail.com

## Abstract

### Introduction

Double disease burden such as Tuberculosis and Diabetes mellitus comorbidity is evident and on rising especially in high burden settings such as Tanzania. There is limited information about the availability of tuberculosis/diabetes integrated healthcare services in Tanzania. Therefore, this study explored the availability and examined the readiness of healthcare facilities with tuberculosis services to manage diabetes mellitus in Tanzania.

### Methods

We abstracted data from the 2014–2015 Tanzania Service Provision Assessment Survey datasets. The service availability was assessed by calculating the proportion of tuberculosis facilities reported to manage diabetes mellitus. There were four domains; each domain with some indicators for calculating the readiness index. High readiness was considered if the tuberculosis facilities scored at least half (≥50%) of the indicators listed in each of the four domains (staff training and guideline, diagnostics, equipment, and medicines) as is recommended by the World Health Organization-Service Availability and Readiness Assessment manual while low readiness for otherwise.

### Results

Out of 341 healthcare facilities with tuberculosis services included in the current study, 238 (70.0%) reported providing management for diabetes mellitus. The majority of the facilities were dispensaries and clinics 48.1%; publicly owned 72.6%; and located in rural 62.6%. Overall, the readiness of tuberculosis facilities to manage diabetes was low (10.8%). Similarly, the readiness was low based on the domain-specific readiness of trained staff and guidelines.

**Data Availability Statement:** The dataset used in the current analysis can be obtained from DHS Program: https://dhsprogram.com. Any researcher can access the data from DHS databases after

registering at (https://dhsprogram.com/data/new-user-registration.cfm) and get permission to access data at (https://dhsprogram.com/data/available-datasets.cfm). The dataset is specifically available from the "SPA datasets" at the DHS Program for Tanzania 2014-15. The SPA dataset can be accessed at https://dhsprogram.com/methodology/survey/survey-display-401.cfm and downloaded at https://dhsprogram.com/data/dataset/Tanzania_SPA_2014.cfm?flag=1.

**Funding:** The author(s) received no specific funding for this work.

**Competing interests:** The authors have declared that no competing interests exist.

## Conclusion

Although the majority of the healthcare facilities with tuberculosis services had diabetes mellitus services the overall readiness was low. This finding provides a piece of evidence to inform the policymakers in high burden and low resource countries to strengthen the co-management of tuberculosis and diabetes.

## Introduction

Despite the decline of the mortality rate of active Tuberculosis (TB) since 1990, TB remains one of the top ten cause of mortality worldwide [1]. As in the year 2015, there were 10.4 million incident TB cases globally [1]. Moreover, in low- and middle-income countries (LMICs) TB is one of the main cause of morbidity and mortality [2]. To tackling the TB burden, the World Health Organization (WHO) through the "End TB Strategy," set a target of reducing TB incidence and mortality at 80% and 90%, respectively by the year 2030 [1, 3].

Although there has been a significant decline of TB incidence in Tanzania from 306/100,000 population in 2015 to 253/100,000 population in 2018, it remains among the 7 TB high burden countries in the world [4].

In contrast to tuberculosis, the prevalence of diabetes mellitus (DM) has been increasing globally since the 1980ˢ. In the year 2014, 422 million adults were living with DM while 3.7 million people died because of DM in the year 2012 [5]. Also, the prevalence has increased in Africa: from 3.1% in the 1980ˢ to 7.1% in 2014 [5]. In 2014, about 12.1 million people in Africa were living with DM and is expected to reach 23.9 million people by 2030 [6]. Although the burden of DM in Africa especially Sub-Saharan Africa (SSA) is in coexistence with other non-communicable diseases (NCDs) [7] the region has also a high burden of Tuberculosis [1].

Tanzania is experiencing a significant burden of NCDs and communicable diseases (CD) [8]. The nationwide survey in 2012 showed that the prevalence of DM among the working-age population (25–64 years) was 9.1% while for a combined DM and pre-diabetes was 20% [9].

The burden of diabetes mellitus can have an impact on achieving the "End TB Strategy" targets by 2030 [10, 11]. Globally, over 1 million people among new patients with tuberculosis had TB/DM comorbidity in the year 2012, [11, 12]. Moreover, studies have shown that there is a significant interaction between diabetes and tuberculosis. DM increases the risk of developing TB as well as poorer disease outcomes [13, 14]. People with DM have a significant three-fold increased risk of active TB compared to the general population [15, 16]. On the other hand, tuberculosis in a patient with diabetes is associated with poorer glycemic control [14, 17]. A case-control study in Tanzania found that the prevalence of diabetes was 16.7% and 9.4% among 803 culture-confirmed pulmonary TB cases and 350 non-TB controls, respectively [18]. In another case-control study in Tanzania found that the prevalence of diabetes was significantly higher among tuberculosis cases compared to controls [19]. Moreover, finding from a prospective cohort study found that TB/DM patients initiated on tuberculosis treatment showed a delay in functional recovery from the disease [20]. Furthermore, in one longitudinal study, it was revealed that diabetes patients with low 25 hydroxyl vitamin D had an increased risk of developing tuberculosis [21].

The increasing dual tuberculosis and diabetes mellitus diagnosis in Tanzania [22] indicate an important public health agenda. However, as of 2015, the National Tuberculosis and Leprosy Program (NTLP) lacked clear commends on screening for DM among TB patients and vice versa [23]. Similarly, the Tanzania National Standard Treatment Guidelines has no information regarding screening for TB among DM patients [24]. Although one study found that

DM health facilities had a low readiness to manage TB [25] there is a paucity of information about the capacity of tuberculosis facilities to manage diabetes mellitus. Knowing both scenarios can help to decide either to co-manage TB/DM in tuberculosis facilities or diabetes facilities. Therefore, this study aimed at assessing the availability and readiness of TB health facilities to manage DM in Tanzania.

## Material and methods

### Study design and data source

The current study was based on the secondary data analysis of the cross-sectional survey of the 2014–2015 Tanzania Service Provision Assessment (TSPA). The TSPA survey was carried out by the Tanzania Ministry of Health, Community Development, Gender, Elderly and Children (MoHCDEC) with technical support from ICF International under the DHS Program. The informed consent was requested and obtained from the manager, the person-in-charge of the facility, or the most senior health worker responsible for client services who was present at the facility. All relevant aspects of the study, including its aim and interview procedures were explained clearly to the respondents before interviews. Those respondents agreed their facilities to participate in the study, provided a signed written informed consent. Therefore, the ethical approval for the current study analysis was automatically deemed unnecessary. Moreover, the permission to use the datasets was obtained from the Demographic Health Survey Program accessed at https://dhsprogram.com/data/new-user-registration.cfm".

The 2014–2015 TSPA survey was designed to collect important information about the availability and readiness of the basic health care services of Tanzanian health facilities. The main purpose was to assess the presence and function of components essential for quality services delivery in the following areas; NCDs, Family Planning (FP), Antenatal care (ANC), Child Health, Maternal, and Newborn Care, Sexually Transmitted Infections, Malaria, tuberculosis, and HIV/AIDS [26].

### Sampling and sample size

The TSPA survey involved a sample of all formal-sector health facilities in Tanzania. A master list of health facilities that consisted of 7,102 verified (active) health facilities in Tanzania was obtained from the MoHCDEC in the Tanzania Mainland and the Ministry of Health (MOH) in Zanzibar. The list included hospitals, health centres, dispensaries, and clinics. These facilities were managed by the public (government, army, and parastatal), and non-public (private-for-profit, and faith-based entities). Four main types of data collection tools were used in the TSPA survey; Facility inventory questionnaire, Health Provider Interview questionnaire, Observation Protocols ANC, FP, and services for sick children, and Exit Interview questionnaires for ANC and FP clients and for caretakers of sick children whose consultations were observed. A total of 1200 health facilities were randomly selected for the TSPA survey. The sample size was designed to offer nationally representative results according to the type of facility, managing authority, and regions for both Tanzania Mainland and Zanzibar [26].

In the current study, we abstracted data from the Facility Inventory file hence the unit of analysis remained at the facility level. The main inclusion criteria were; the health facility must have agreed to participate in the survey, being opened during the day of the interview, and providing tuberculosis services. A total of 1200 health facilities across Tanzania were selected to participate in the 2014–2015 TSPA survey. Of the 1200 facilities, seven refused to participate, four were closed on the day of the survey and, 847 were not providing TB services hence were excluded. Therefore, a final sample of 341 health facilities was included in the current study analysis. For more details, refer to Fig 1.

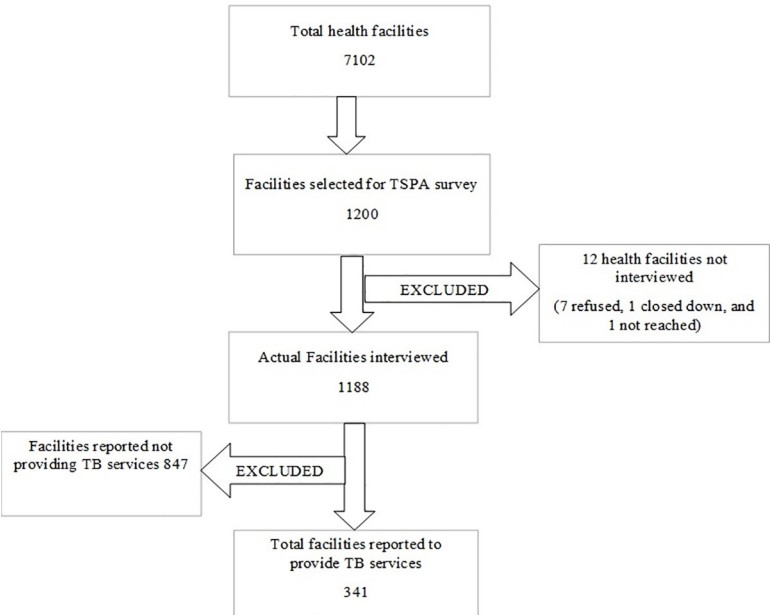

**Fig 1. Selection of health facilities included in the current analysis.**

## Operational definition of terms

**Service availability.** In this study, it is defined as the "percentage of healthcare facilities with tuberculosis services offering diabetes mellitus services": diagnosis and treatment. In this study TB services means both diagnosis and treatment.

**Readiness.** In this study, it is defined as "the capacity of the healthcare facilities with tuberculosis services to provide management for diabetes mellitus". The tuberculosis facility was considered to have "high readiness" if they scored at least a half ($\geq$50%) of the indicators in each of the four domains suggested by the World Health Organization-Service Availability and Readiness Assessment (WHO-SARA) reference manual. The domains were; staff training and guideline, diagnostics (glucometer, urine dipstick for glucose, and urine dipstick for protein), equipment (blood pressure machines, stethoscopes, weighing scale, and Stadiometer), and medicine (Metformin, Glibenclamide, Insulin injection, and glucose solution). In contrast, the facility was considered to have "low readiness" if they scored less than half (<50%) of the aforementioned indicators in each of the four domains.

**Clinic.** In this study, it refers to a non-publicly owned health facility that provides outpatient curative services [27]. The clinic usually is a standalone health facility different from diseases specific clinics within hospitals.

**Dispensary.** Is regarded as the lowest level health facility that provides only outpatient curative services just like clinics [27]. It can be publicly or non-publicly owned.

**Health centres.** Constitutes the first referral health facility that provides a wide range of services including basic curative and operative services [27]. Also, it can be publicly or non-publicly owned

## Measure of variables

**Outcome variables.** The first outcome variable was the "availability" of diabetes services in tuberculosis health facilities. The availability was calculated as a percentage of tuberculosis facilities providing both diagnosis and treatment of diabetes.

The second outcome variable was the "readiness" of the TB health facilities. The readiness variable was rated as an index categorized into four domains as suggested by the WHO-SARA reference manual specific for diabetes services [28]. The first domain was staff and guidelines which had two indicators; the presence of a guideline for DM management, and at least one staff who received refresher training in DM diagnosis and treatment. The facilities with guidelines were categorized as "Yes" and "No" for otherwise. Similarly, facilities with at least one staff member who had received refresher training in DM diagnosis and treatment within 24 months were categorized as "Yes" and "No" for otherwise. The second domain was equipment which had five indicators; digital and manual blood pressure machines, stethoscopes, adult weighing scale, and Stadiometer. The facilities with the five aforementioned indicators were categorized as "Yes" and "No" for otherwise. The third domain was diagnostics which had three indicators; glucometer, urine dipstick for glucose, and urine dipstick for protein. The facilities with the three aforementioned indicators were categorized as "Yes", otherwise were categorized as "No". The fourth domain was basic medicine which had four indicators: the availability of the four basic DM medication in the Tanzania context; metformin, Glibenclamide, insulin injection, and glues solution. The facilities with the four DM-drug were categorized as "Yes", otherwise were categorized as "No". The responses were aggregated into an index score to calculate a composite score as per the WHO-SARA reference manual [28]. The index score was calculated by adding the presence of each indicator, with equal weight given to each of the domains and each of the indicators within the domains. Since the target was 100%, each domain accounted for 25% (100%/4) of the index. The percentage for each indicator within the domain was equal to 25% divided by the number of indicators within that domain. The overall facility readiness was then calculated as the average of domain indices. The tuberculosis facility that scores at least half (equivalent to the median value of 12.5% and above) in each domain and adding up to the overall of 50% or more were considered to have "high readiness" for the management of TB while those with less than 50% were considered to have "low readiness." The cut-off point used in this study was also used in the previous studies to dichotomize the outcome variable [25, 29–33].

**Explanatory variables.** The facility type was categorized as "hospital," "health centre," and "clinic/dispensary"; the residence was categorized as "urban" and "rural", and managing authority was categorized as "public (government, army, and parastatal)" and "non-public (private-for-profit, and faith-based entities.)"

## Statistical analysis

SPSS version 22 (SPSS, Chicago, IL) was used for data analyses. We weighted all the estimates to correct for non-responses and disproportionate sampling. The results of the current study are presented using descriptive statistics.

## Results

### Baseline characteristics of tuberculosis health facilities

A total of 341 health facilities reported to provide management for tuberculosis were assessed. Of these, 200 (58.8%) were dispensaries and clinics, 265 (77.6%) were publicly owned, and 220 (64.5%) were rurally located (Table 1).

### Availability of diabetes services

A total of 341 facilities reported to provide tuberculosis management were assessed for the availability of diabetes services. Overall, about two-thirds 238 (70.0%) of tuberculosis facilities

**Table 1. Distribution characteristics of facilities reported to provide tuberculosis services [N = 341].**

| Characteristics | Facilities managing TB n [%] |
|---|---|
| | Yes |
| **Facility type** | |
| Hospital | 42 [12.2] |
| Health centers | 99 [29.0] |
| Dispensaries and Clinics | 200 [58.8] |
| **Managing authority** | |
| Public | 265 [77.6] |
| Non-Public | 76 [22.4] |
| **Residence** | |
| Urban | 121 [35.5] |
| Rural | 220 [64.5] |
| Total | 341 [100.0] |

reported providing management of diabetes mellitus (Fig 2). The majority of the health facilities were dispensaries and clinics 114 (48.1%), publicly owned 173 (72.7%), and rurally located 149 (62.6%) (Fig 3).

## The readiness of tuberculosis facilities to provide diabetes management

The overall readiness of tuberculosis facilities to provide diabetes management was low, 10.8%. Moreover, when assessed by facility characteristics (facility type, managing authority, and facility residence), none of the facility was ready to provide management of DM (Fig 4).

## Domain-specific readiness index

None of the tuberculosis facilities was ready to provide management for tuberculosis in terms of trained staff and guidelines domain. However, all facility characteristics were ready in terms of equipment (readiness index ≥ 12.5%). In terms of diagnostics domain; hospitals, health

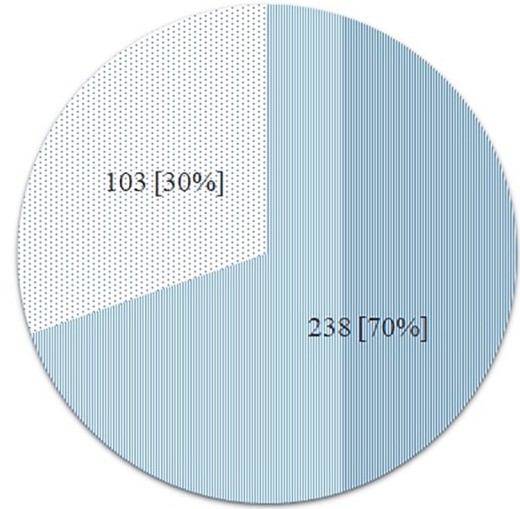

**Fig 2. Percentage of tuberculosis facilities with available diabetes services [N = 341].**

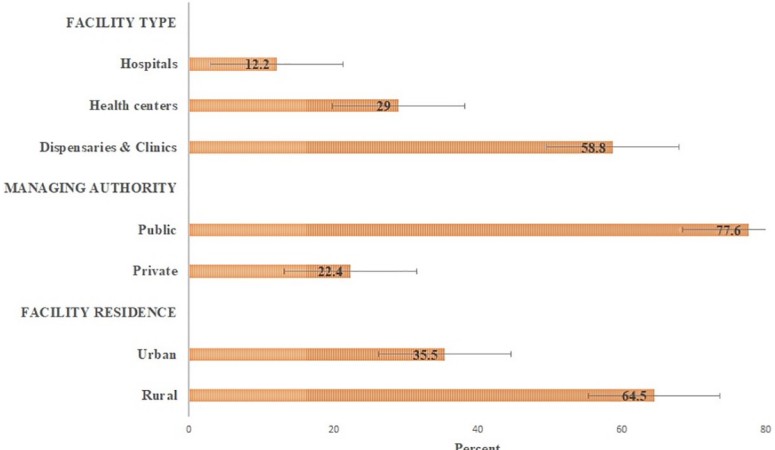

**Fig 3. Distribution characteristics of tuberculosis facilities reported to provide diabetes mellitus services [N = 238].** Error bar represents standard error.

centres, non-public owned, and urban located facilities were ready. Regarding the medicine domain; only hospitals and non-public owned facilities were ready (readiness index $\geq$ 12.5%) to provide management of diabetes (Table 2).

## Discussion

Integrated management of tuberculosis-diabetes in health facilities dedicated to providing services for either of the two diseases condition is of paramount importance in a setting with a high burden of tuberculosis and diabetes such as Tanzania. This study assessed the availability and readiness of Tanzanian tuberculosis health facilities to provide management for diabetes mellitus.

In this study, a large proportion of tuberculosis health facilities reported providing management for diabetes mellitus. Lower-level health facilities (health centres, dispensaries and clinics); public health facilities, and rural health facilities constituted a large proportion compared to hospitals, non-public health facilities, and urban health facilities, respectively. This finding reflects the fact that in Tanzania, there are more public health facilities than non-public health

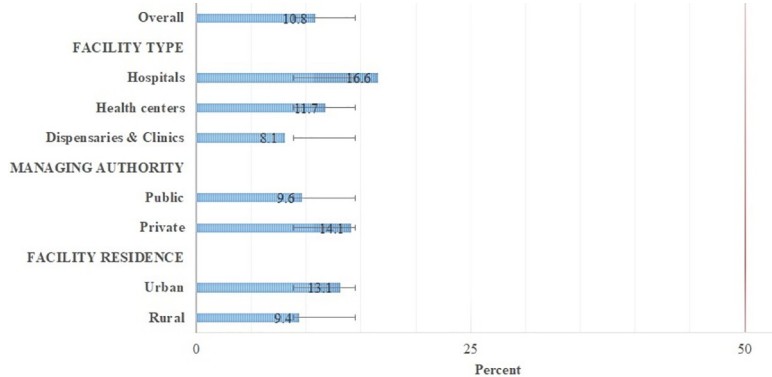

**Fig 4. Overall readiness index by facility characteristics (N = 238).** The red line depicts the cut off below which facility was regarded as having low readiness. Error bar represents the standard deviation from the mean readiness index.

**Table 2. Domain-specific readiness for DM management by facility characteristics (N = 238).**

| Variables | Facility type | | | Managing authority | | Facility location | |
|---|---|---|---|---|---|---|---|
| | Hospital % (n = 41) | Health centers % (n = 83) | Disp. & Clinics % (n = 114) | Govern. % (n = 173) | Non-public % (n = 65) | Urban % (n = 89) | Rural % (n = 149) |
| **Staff and guidelines** | | | | | | | |
| Staff trained in DM Yes | 0.7 | 1.2 | 2.0 | 1.0 | 2.8 | 1.9 | 1.3 |
| Guidelines for DM Yes | 10.9 | 7.9 | 5.0 | 7.6 | 5.5 | 7.4 | 6.9 |
| **Mean domain score** | 5.8 | 4.6 | 3.5 | 4.3 | 4.2 | 4.7 | 4.1 |
| **Equipment** | | | | | | | |
| Digital BP machine available—Yes | 11.1 | 9 | 6.7 | 6.5 | 12.9 | 11.2 | 6.5 |
| Manual BP machine available—Yes | 21.8 | 21.0 | 22.7 | 21.9 | 22.1 | 21.1 | 22.5 |
| Stethoscope available Yes | 24.2 | 24.2 | 23.9 | 24.2 | 23.7 | 23.7 | 24.2 |
| Adult weighing scale available—Yes | 23.6 | 20.9 | 20.8 | 21.3 | 21.3 | 20.4 | 21.8 |
| Stadiometer available Yes | 17.6 | 13.5 | 13.2 | 14.4 | 13.2 | 15.5 | 13.2 |
| **Mean domain score** | 19.7 | 17.7 | 17.5 | 17.7 | 18.6 | 18.4 | 17.6 |
| **Diagnostics** | | | | | | | |
| Glucometer available Yes | 20.8 | 17.2 | 7.5 | 10.4 | 20.8 | 17.4 | 10.8 |
| Urine dipstick for protein available—Yes | 22.0 | 18.1 | 9.2 | 12.2 | 20.8 | 18.6 | 12.1 |
| Urine dipstick for glucose available—Yes | 22.0 | 17.8 | 8.8 | 11.4 | 21.2 | 18.0 | 11.8 |
| **Mean domain score** | 21.6 | 17.7 | 8.5 | 11.3 | 20.9 | 18.0 | 11.6 |
| **Medicines** | | | | | | | |
| Metformin available Yes | 20.6 | 8.9 | 4.8 | 5.5 | 18.1 | 14.7 | 5.5 |
| Glibenclamide available Yes | 19.5 | 8.3 | 2.5 | 4.7 | 14.7 | 12.5 | 4.4 |
| Insulin available Yes | 20.7 | 4.0 | 0.4 | 3.5 | 9.3 | 9.3 | 2.6 |
| Glucose solution available—Yes | 16.2 | 6.5 | 3.2 | 5.7 | 9.0 | 9.0 | 5.1 |
| **Mean domain score** | 19.3 | 6.9 | 2.7 | 4.9 | 12.8 | 11.4 | 4.4 |

facilities [34]. Before 2007, TB services were restricted to public health facilities and very few non-public facilities [35, 36]. Also, there has been an increasing number of non-public health facilities involved in providing tuberculosis services in Tanzania [23]. Moreover, rural health facilities constituted a large proportion because in rural there are more health facilities compared to urban [26]. Also, over two-thirds of the population of Tanzania is predominantly rural [37]. The increasing burden of tuberculosis [1, 4] and diabetes mellitus [9, 38] as well as the high rate of dual TB/DM diagnosis [18–21] in Tanzania, is an urgent call for a government to establish a wide coverage of DM services in facilities dedicated to managing TB. This can help to reduce the incidence of active TB, which is increasing in Tanzania [4], as well as achieving the "End TB Strategy." by 2030.

The WHO and International Union Against Tuberculosis and Lung Diseases established a collaborative framework for care and control of TB and Diabetes. The framework recommends national programmes, clinicians, and other people engaged in the prevention and control of TB and DM to establish a collaborative response to both diseases [39]. However, the current study found that TB facilities were not ready to provide management of diabetes based on all four readiness domains (i.e., trained staff and guidelines, equipment, diagnostics, and basic medicines). This low readiness can be attributed to an overburdened healthcare workforce [40]. The low readiness in this study is comparable to a study in South Africa whereby bidirectional screening and co-management of TB and DB was found to be weak [40]. It is necessary for the government in collaboration with the non-public sector to design and plan measures to address this low readiness. This will help to mitigate an increasing incidence of tuberculosis as well as TB/DM comorbidity.

Furthermore, the current study found that hospitals and non-public health facilities were ready to provide management of diabetes mellitus in terms of diagnostics and medicine domains compared to dispensaries and clinics, and public health facilities. Although non-public health facilities constitute less than one-third of the health facilities they have sustainable, efficient, and good quality healthcare services compared to public health facilities [41, 42]. On the other hand, public health facilities have a long process involving multiple levels of authorization in ordering and procurement of medicine and equipment [43, 44]. Moreover, inadequate funding, poor planning, inadequate monitoring and evaluation, and an overburdened workforce more especially in lower-health facilities affect the quality of healthcare and performance of service provision [45]. These reasons can have explained why public health facilities and lower-level health facilities were not ready for management of DM.

Having healthcare providers with updated knowledge and skills as well as the availability of specific disease management guidelines are particularly important for a facility readiness to manage a particular disease condition. In the current study, none of the tuberculosis facilities was ready to manage diabetes in terms of trained staff and guidelines. Despite an increase in the healthcare workforce, there has been unequal deployment resulting in shortages, especially in lower-health facilities. Also, limited funding and budgetary shortfall affect the procurement of medical supplies [45].

The current study used a nationally representative sample of health facilities hence the findings can be generalized to other health facilities across Tanzania. However, the current study has some limitations. This was a cross-sectional design: the outcome and exposure were assessed simultaneously hence the findings may not reflect the change in services over time. Also, this study did not assess the availability of other important DM diagnostic methods such as glycosylated haemoglobin (HbA1c) assay. This is an important diagnostic guide for monitoring glycemic control. However, in Tanzania, the HbA1c assay machines are very few in most of the health facilities.

## Conclusions

Although the majority of the tuberculosis healthcare facilities reported to provide management for diabetes mellitus the overall readiness was low. This finding provides baseline evidence to inform the policymakers in high burden and low resource countries to consider strengthening the co-management of tuberculosis and diabetes. Investment should focus particularly on increasing funding, healthcare workforce, and in-service training.

## Acknowledgments

We gratefully acknowledge the ICF International, Rockville, Maryland, USA, through DHS programme for permitting us to access the Tanzania SPA 2014–2015 dataset.

## Author Contributions

**Conceptualization:** Festo K. Shayo, Sigfrid Casmir Shayo.

**Data curation:** Festo K. Shayo.

**Formal analysis:** Festo K. Shayo.

**Software:** Festo K. Shayo.

**Writing – original draft:** Festo K. Shayo, Sigfrid Casmir Shayo.

**Writing – review & editing:** Festo K. Shayo, Sigfrid Casmir Shayo.

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
