## [Decision Letter · Decision Letter 0]

15 Apr 2021

PONE-D-20-33227

Readiness of Tuberculosis healthcare facilities to manage Diabetes mellitus in Tanzania: a nationwide analysis for evidence-informed policy-making in high burden settings

PLOS ONE

Dear Dr. Shayo,

Thank you for submitting your manuscript to PLOS ONE. After careful consideration, we feel that it has merit but does not fully meet PLOS ONE’s publication criteria as it currently stands. Therefore, we invite you to submit a revised version of the manuscript that addresses the points raised during the review process.

We look forward to receiving your revised manuscript.

Kind regards,

Joel Msafiri Francis, MD, MS, PhD

Academic Editor

PLOS ONE

Journal Requirements:

Reviewers' comments:

Reviewer's Responses to Questions

**Comments to the Author**

1. Is the manuscript technically sound, and do the data support the conclusions?

Reviewer #1: Yes

Reviewer #2: Yes

2. Has the statistical analysis been performed appropriately and rigorously? 

Reviewer #1: Yes

Reviewer #2: Yes

3. Have the authors made all data underlying the findings in their manuscript fully available?

Reviewer #1: Yes

Reviewer #2: Yes

4. Is the manuscript presented in an intelligible fashion and written in standard English?

Reviewer #1: Yes

Reviewer #2: Yes

5. Review Comments to the Author

Reviewer #1: The paper is well written and meet the above criteria. However few questions need to be answered by the author. Also need some grammar check. Few comments for the author to respond is attached with this document

Reviewer #2: The manuscript is well written. The manuscript describes an important aspect in the management of Diabetes Mellitus and Tuberculosis. The authors have successfully applied statistical methods suitable to answer the proposed research questions.

6. PLOS authors have the option to publish the peer review history of their article (what does this mean?). If published, this will include your full peer review and any attached files.

Reviewer #1: No

Reviewer #2: No

---

## [Author Response · Author response to Decision Letter 0]

9 Jun 2021

Responses to Reviewers` comments 

Reviewer 1: 

General comment:

The study is well written

Response 

Thank you very much for taking the time to review our manuscripts. We appreciate your comments, recommendations and questions which have helped us to improve our manuscript as well for future write-up. 

Specific comments 

Comment 1

Line number 77, there has been an increased burden of TB-DM comorbidity the reference cited here reference 19 does not indicate that there is an increase in the burden of TB and DM, also reference number 20. May be the author is referring other studies, the author should cite the correct citation for this statement

Response 

In response to the reviewer`s comments, the references have been revised and the respective section paraphrased to reflect the cited articles (Page 4 line 82). The background section has been revised hence some cited references sequence were changed.

Comment 2

Line 99-104

Sputum smear microscopy remains the gold standard for-----------------].

All these have nothing to do with the readiness of the TB facilities to provide DM care? So this is irrelevant in this introduction

Response 

According to the reviewer`s suggestion, the section has been deleted. 

Comment 3

Line 114 and 115. As the burden of TB/DM comorbidity is on rising in Tanzania there is a dearth of information about the capacity of tuberculosis facilities to manage diabetes.

Provide reference for this information

Response 

According to the reviewer`s comment, the reference has been inserted (Page 4 line 82). 

Comment 4

Study design is missing

Response

In response to the reviewer`s comment, the information about study design has been added (Page5 line 96).

Comment 5

Line 132, NCDs, where the survey able to establish weather or not the health facilities is read to deliver services for all NCDs or there were few selected NCDs like the selected communicable disease? 

Response

According to the reviewer`s questions, not all NCDs were selected. The survey selected only the major four NCDs: Chronic Respiratory Diseases, Cardiovascular Diseases, Hypertension, and Diabetes Mellitus. The analyses of these NCDs have already published. Kindly see some of the articles below: 

1. Are Tanzanian health facilities ready to provide management of chronic respiratory diseases? An analysis of national survey for policy implications. PloS one. 2019 Jan 7;14(1):e0210350.

2. Disparities in availability of services and prediction of the readiness of primary healthcare to manage diabetes in Tanzania. Primary care diabetes. 2021 Apr 1;15(2):365-71.

3. Preparedness of lower-level health facilities and the associated factors for the outpatient primary care of hypertension: Evidence from Tanzanian national survey. PloS one. 2018 Feb 15;13(2):e0192942.

Comment 6

Line 168-170 The domains were; staff training and guideline, diagnostics (sputum smear microscopy), equipment (blood pressure machines, stethoscopes, weighing scale, and Stadiometer),

None of these mentioned here are used to diagnose DM. How DM was diagnosed. If the mentioned survey uses this tool to conclude that they diagnose DM it is wrong

Response

In response to the reviewer`s comment, the section has been revised to include the diagnostic tools for Diabetes Mellitus assessed in that survey (Page 7 lines 141-142).

Regarding the equipment (blood pressure machines, stethoscopes, weighing scale, and Stadiometer) are used for disease monitoring and detection of complications. For instance, a blood pressure machine and stethoscope is for detecting hypertension and cardiorespiratory complications, respectively. The weighing scale and Stadiometer are for assessing body weight and body mass index in respect to cardio-metabolic parameters such as LDL cholesterol. 

Comment 7

Table 1. When categorizing the Health facilities there is only public and Private, where are the FBO facilities? These are neither public nor private you need to categorize this

Response

According to the reviewer`s questions, the intention was to compare the public facilities vs. non-public facilities. Therefore, to make it clear for the readers, the term private has been replaced with "non-public" to include all other non-public facilities (Page 10 line 208). Also, it has been defined on Page 6 line 112; Page 10 lines 19-194). 

Comment 8

Line 233 when referring Clinic does this mean stand alone clinic or TB clinic located in a Dispensary, health center or Hospital? Please define this

Response

According to the reviewer`s question, in this study, a clinic refers to a non-publicly owned health facility that provides outpatient curative services. Usually, it is a standalone health facility different from diseases specific clinics within hospitals. The operational definition has been added on Page 7 lines 146-148.

Comment 9

When it comes to TB facilities there are two types according to Tanzania National TB program. There are TB facilities which offer both TB diagnosis and treatment (DOT) Services and other TB clinic which offer only treatment and this are referred to as DOTS centre Indicate which one are referred here

Response

In response to the reviewer`s comments and question, the TB facilities referred in this study were those providing both diagnosis and treatment (Page 7 line 134-135). 

Comment 10

Line 270-272 Based on facility characteristics, the majority of TB facilities with the availability of DM services were health centres and dispensaries, publicly owned, and rural located facilities. The author need to revisit the analysis section in Tanzania all hospitals have the basic facilities for diagnosis of DM, when saying majority of the facilities with DM diagnosis are HC and dispensary is because HC and DM were more than Hospitals in this analysis.

Response

In response to the reviewer`s comments and suggestions, the section has been revised and correction made (Page 13 lines 251-252, and page 14, lines 253-254). 

Comment 11

Page 15 last paragraph starting with line 288, the author need to explore more why the TB clinic are not ready to provide DM services/ In most cases TB facilities are under staffed just as the whole Health sector in Tanzania is under staffed, adding more burden of managing DM the staff will not be comfortable?

Response

In response to the reviewer`s comments and question, the explanation has been provided on Page 14 lines 273-274 and page 15 lines 275-276, 284-290; and page 16 lines 296-297. 

Reviewer 2: 

General comment:

The study addressed a very important aspect currently facing developing World; coexistence of a longtime infectious disease like Tuberculosis (TB) together with an emerging challenge of Non-Communicable disease like Diabetes (DM). 

Response 

Thank you very much for taking the time to review our manuscripts. We appreciate your comments, recommendations and questions which have helped us to improve our manuscript as well for future write-up. 

Specific comments 

Comment 1

Include at least a line on the aim/objective(s) of the study

Response 

In response to the reviewer`s suggestion, the aim/objective has been included (Page 1 lines 19-20).

Comment 2

First line of the conclusion: from your result section, none of the health facilities was ready for managing TB-DM coexistence. Hence the readiness was not low, but it was not there at al.

Response 

According to the reviewer`s comments, the section has been revised for clarity (Page 2, lines 34-35 and 37-38). The conclusion is based on overall readiness while the sentence on the results included specific domain readiness. 

Comment 3

The introduction is too long for a typical scientific manuscript. Please consider including the following: Burden of TB, burden of DM, interaction of TB and DM, what is currently done/known in Tanzania regarding TB-DM management, what are the gaps intended to be filled by the proposed study and what is proposed to be done in the study.

Response 

In response to the reviewer`s comment and suggestions, the introduction has been revised and shortened (Page 3-5, lines 44-91).

Comment 4

Results: Line 223: I think you should clearly say something like “Majority of the health facilities were dispensaries and clinics (xx%)………………………………………

Response

In response to the reviewer`s suggestion, the sentence has been revised (Page 11, lines 213-215).

Comment 5

Discussion: The authors did a good job to discuss the implications of the findings. However, the authors need to review the discussion section to makes sure that there are repetitions of what has been reported in the result section. The discussion needs to tell the readers the interpretations of the findings.

Moreover, the authors did not tell the readers what other articles found about the same issues addressed in the article under discussion. The authors have the obligation to compare and contrast with other articles and discuss whatever they find in other articles.

Response

According to the reviewer`s comments and suggestions, there is only one related study found to compared and contrast with the current study (Page 14 and 15, lines 274-276). Moreover, the discussion has been revised to omit repetition of the results (Page 13, 14, 15 and 16; lines 251-261 and 267-297.

Comment 6

Conclusion: The conclusion almost repeated result section. Please revise the conclusion to clearly state what is concluded as a result of the findings and the discussion of this article.

Response

In response to the reviewer`s comments and suggestions, the conclusion has been revised (Page 16, lines 307-311).

---

## [Decision Letter · Decision Letter 1]

25 Jun 2021

Readiness of Tuberculosis healthcare facilities to manage Diabetes mellitus in Tanzania: a nationwide analysis for evidence-informed policy-making in high burden settings

PONE-D-20-33227R1

Dear Dr. Shayo,

We’re pleased to inform you that your manuscript has been judged scientifically suitable for publication and will be formally accepted for publication once it meets all outstanding technical requirements.

Kind regards,

Joel Msafiri Francis

Academic Editor

PLOS ONE

Additional Editor Comments (optional):

Address the additional minor discretionary comments below.

Title: Readiness of Tuberculosis healthcare facilities to manage Diabetes mellitus in Tanzania: a nationwide analysis for evidence-informed policy-making in high burden settings

Abstract: Consider using “health facilities with TB treatment service rather than TB facilities (It should also be clear if TB facilities referred to health facilities with TB treatment or TB diagnosis?)

Reviewers' comments:

Reviewer's Responses to Questions

**Comments to the Author**

1. If the authors have adequately addressed your comments raised in a previous round of review and you feel that this manuscript is now acceptable for publication, you may indicate that here to bypass the “Comments to the Author” section, enter your conflict of interest statement in the “Confidential to Editor” section, and submit your "Accept" recommendation.

Reviewer #2: All comments have been addressed

2. Is the manuscript technically sound, and do the data support the conclusions?

Reviewer #2: Yes

3. Has the statistical analysis been performed appropriately and rigorously? 

Reviewer #2: Yes

4. Have the authors made all data underlying the findings in their manuscript fully available?

Reviewer #2: Yes

5. Is the manuscript presented in an intelligible fashion and written in standard English?

Reviewer #2: (No Response)

6. Review Comments to the Author

Reviewer #2: The manuscript discussed a very important topic in health. There is a growing threat regarding Diabetes Mellitus Worldwide. Integration of service is an important aspect in disease control.

7. PLOS authors have the option to publish the peer review history of their article (what does this mean?). If published, this will include your full peer review and any attached files.

Reviewer #2: No

---

## [Editor Report · Acceptance letter]

2 Jul 2021

PONE-D-20-33227R1 

Readiness of Healthcare facilities with Tuberculosis services to manage Diabetes mellitus in Tanzania: a nationwide analysis for evidence-informed policy-making in high burden settings 

Dear Dr. Shayo:

I'm pleased to inform you that your manuscript has been deemed suitable for publication in PLOS ONE. Congratulations! Your manuscript is now with our production department. 

Kind regards, 

on behalf of

Dr. Joel Msafiri Francis 

Academic Editor

PLOS ONE